# Magnetic Measurement of Zn Layer Heterogeneity on the Flange of the Steel Road Barrier

**DOI:** 10.3390/ma15051898

**Published:** 2022-03-03

**Authors:** Martin Pitoňák, Ján Ondruš, Peter Minárik, Tibor Kubjatko, Miroslav Neslušan

**Affiliations:** 1Faculty of Civil Engineering, University of Žilina, Univerzitná 1, 01026 Zilina, Slovakia; 2The Faculty of Operation and Economics of Transport and Communications, University of Žilina, Univerzitná 1, 01026 Zilina, Slovakia; jan.ondrus@fpedas.uniza.sk; 3Faculty of Mathematics and Physics, Charles University, Ke Karlovu 5, 121 16 Prague 2, Czech Republic; peter.minarik@mff.cuni.cz; 4Institute of Forensic Research and Education, University of Žilina, Univerzitná 1, 01026 Zilina, Slovakia; tibor.kubjatko@usi.sk; 5Faculty of Mechanical Engineering, University of Žilina, Univerzitná 1, 01026 Zilina, Slovakia; miroslav.neslusan@fstroj.uniza.sk

**Keywords:** Barkhausen noise, Zn layer, steel barrier

## Abstract

This study deals with monitoring of Zn layer heterogeneity on the flange of steel road barriers using magnetic measurements. The Barkhausen noise technique is employed for such purpose, and parameters extracted from Barkhausen noise signals are correlated with the true thickness of the Zn layer. The true values of the Zn layer were obtained from the metallographic images, as well as the thickness gauge CM-8825FN (Guangzhou Landtek Instruments Co. Ltd., Guangzhou, China) device. It was observed that the diffusion region lies below the Zn protective layer, which makes the thickness of the Zn layer obtained from the CM-8825FN device thicker than that measured on the metallographic images. For this reason, the chemical gradient of Zn below the Zn layer can be reported, and it affects Barkhausen noise emission. Barkhausen noise decreases along with increasing thickness of the Zn layer, and Barkhausen noise envelopes are shifted to stronger magnetic fields. The number of strong MBN pulses drops down with the increasing thickness of Zn coating at the expense of the increasing number of the weak MBN pulses. The thickness of Zn coating can be polluted by the solidification of Zn melt after galvanizing. The presence of the diffusion layer dims the contrast between ferromagnetic and paramagnetic phases.

## 1. Introduction

Steel road barriers represent a critical component in the safety of road traffic. The energy which the barriers have to absorb during collisions are clearly stated in the national standards [1]. Operation of steel road barriers depends on many aspects, such as their installation, maintenance, cross sectional profile, matrix properties, angle of impact, etc. [1,2,3]. One of the critical aspects in their state can also be expressed in terms of corrosion attack. To slow the corrosion damage of steel road barriers, these components are subjected to the Zn coating (galvanising) in order to generate a protective layer on their surface [4]. The thickness of Zn on the components can vary as a result of the shape effect (for example, cross-sectional profile of the barrier), conditions during galvanizing, manner in which galvanizing is carried out, etc. [4]. The thickness of the Zn layer is one of the parameters strongly affecting the duration of corrosion protection since the Zn layer tends to dissolve in time [5]. For this reason, it is expected that the evolution of corrosion attack and the corresponding corrosion damage can vary with respect to the different regions of steel road barrier and/or among the different barriers.

The Zn galvanizing process is widely investigated, and new regimes with respect to galvanizing temperature or/and chemistry of bath are proposed with respect of their functionality. Kania et al. [6] reported the negative influence of Bi added to a Zn bath on the corrosion resistance due to the presence of Bi precipitates. Li et al. [7] remarkably improved the adhesion of Zn coating after the cold galvanizing by the use of γ-chloropropyl triethoxysilane. Cetinkaya et al. [8] reported the impact of chemical treatment on Zn-Al-Mg coating produced by a hot-dip process in order to increase Zn fraction at the coating surface. The influence of galvanizing temperature on corrosion resistance was also investigated [9], as well as the contribution of Mn to the morphology and corrosion resistance of Zn coating [10]. Finally, Kang et al. [11] investigated Fe-Zn reaction during a hot-dip galvanizing and its influence on microcracking. The aforementioned brief list of the latest findings in this field indicates that Zn coating technology is widely investigated, and many improvements have been proposed in order to improve the corrosion resistance of bodies. However, a fast, reliable, and especially non-destructive technique (for their monitoring) being sensitive to the thickness of the Zn layer could also be beneficial.

Magnetic Barkhausen noise (MBN) has already been reported as a suitable technique for similar purposes [12,13]. It was reported that MBN can provide information about the Zn galvanizing process of wires [12]. This technique was also employed for monitoring coating heterogeneity in the case of steel S460MC [13]. Čillikova et al. [14] also demonstrated that the MBN technique is sensitive to the coating generated under the variable regimes of heat treatment, and MBN is a function of matrix softening during annealing. Santa-aho [15] reported a decrease of MBN when the thin layer of non-ferromagnetic nitrides can be found on the surface. The nitriding process produces the thin nitriding layer and underlying diffusion layer, which decreases MBN [16,17]. MBN decreases along with the increasing thickness of the non-ferromagnetic layer on the surface (and/or hard ferromagnetic) [16] as a result of superimposing effects clearly explained in previous studies [12,17].

The origin of the MBN effect can be found in the discontinuous motion of domain walls (DWs) when the external magnetic field is altered [18,19]. Electromagnetic pulses produced during such motion can be detected on the free surface and contain complex information about the stress state and/or microstructure of a ferromagnetic matrix expressed in a variety of terms [20,21,22,23]. The sensing depth of MBN technique is also important in the studies in which the near-surface layer is altered, or the coating varies in thickness [24,25]. This study reports the heterogeneity of the Zn layer on the flange of a steel road barrier and attenuation of MBN signals resulting from the propagation of electromagnetic pulses towards the free surface. The diversity of MBN parameters is analyzed as a function of variable Zn layer thickness. As compared with the previous studies [13,26] focused on the presence of coating or corrosion on the ferromagnetic matrix, this study also reports about MBN attenuation as a result of diffusion layer laying below the non-ferromagnetic coating. Zn chemistry gradient at the boundary between Zn coating and the steel matrix dims the contrast between ferromagnetic and paramagnetic phases. The contribution of the Zn melt solidification after galvanizing is investigated as well.

## 2. Experimental Conditions

Investigation of the Zn layer thickness was carried out in 7 regions of different Zn layer thickness, preliminarily indicated by the coating thickness gauge CM-8825FN. This gauge operates on the basis of magnetic contrast between the non-ferromagnetic coating and the ferromagnetic body. The measured thickness of the Zn layer varied from 106 up to 235 μm. Measurements were carried out on the flange of new (unused) steel road barrier NH4 of wall thickness 3 mm, length of 4 m, and profile as that depicted in Figure 1. The microstructure of the steel matrix is composed of ferrite grains with small regions of pearlite, see Figure 2. The chemical composition of the steel can be found in Table 1. Elongation at break 33 ± 2%, yield strength 330 ± 15 MPa, and ultimate strength 415 ± 18 MPa were obtained from the uniaxial tensile test carried out on the Instron 5985 device (samples were cut and loaded along the direction of the flange).

The profile of the flange was produced by cold rolling. In order to obtain a Zn layer on the flange surface, the hot-dip galvanizing process in a bath of molten Zn was carried out at an elevated temperature of about 450 °C. Further conditions are not known.

MBN was measured by RollScan 350 (Stresstech, Jyväskylä, Finland) and analyzed with MicroScan 600 software (Stresstech, Jyväskylä, Finland) (magnetizing voltage 16 V, magnetizing frequency 125 Hz, sensor type S1-18-12-01, frequency range of MBN pulses in the range 10–1000 kHz, 10 bursts, sampling frequency 6.4 MHz). The raw MBN signals were post-processed in MicroScan 600 software, and MBN features were extracted as follows:−effective (rms) value of the signal indicated as MBN,−shape and height of MBN envelopes,−*PP* (Peak Position) refers to the position in a magnetic field in which an envelope attains the maximum,−number of MBN pulses and their distribution.

Due to magnetic anisotropy of the produced flange, the MBN measurements were carried out along the barrier length (RD) as well as in the transverse direction (TD), see Figure 1.

To measure the true thickness of the Zn layer, small specimens of length 20 mm were cut from the flange in the measured position along the flange length. These specimens were prepared for light metallograph (LM) (hot molded, ground, polished, and etched by 3% Nital for 5 s). Chemical analysis across the Zn layer was performed with a scanning electron microscope (SEM, AZoNetwork UK Ltd., Manchester, UK) ZEISS Auriga Compact equipped (AZoNetwork UK Ltd., Manchester, UK) with EDAX (energy-dispersive X-ray, AZoNetwork UK Ltd., Manchester, UK) EDX detector (AZoNetwork UK Ltd., Manchester, UK).

In order to investigate the alterations of surface microhardness initiated by cold rolling and the subsequent galvanizing, the HV0.2 microhardness measurements at the different depths were carried out by the use of an Innova Test 400TM (Innovatest, Maastricht, The Netherlands) tester by applying a 200 g force for 10 s on the longitudinal cuts.

## 3. Results of Experiments and Their Discussion

Figure 3 depicts the metallographic images of Zn coating on the surface of the flange. These figures show the typical columnar Zn region of different thicknesses formed during galvanizing at elevated temperatures at the investigated positions initially detected by the thickness gauge CM-8825FN. It is considered that the differences in Zn coating thickness are due to non-homogenous galvanizing conditions within the flange in the bath and/or the heterogeneity of the microstructure of the flange. Figure 3f shows that the structure of Zn layer on the surface is composed of two different regions. The first one represents the usual columnar Zn coating formed during galvanizing as a product of a chemically activated process at elevated temperatures, whereas the second one is a product of the deposition of the solidified Zn melt on the Zn coating after galvanizing (see also Figure 4). Note, the small regions of solidified Zn melt on the Zn coating make the Zn layer on the flange surface much higher. However, it is not known how the solidified Zn melt on the columnar Zn coating contributes to the corrosion protection of a flange. Conversely, these isolated regions can be easily distinguished by visual inspections. Due to the presence of the solidified Zn melt on the columnar Zn region, it is distinguished between the Zn coating (linked with the columnar region only) and the Zn layer (linked with the columnar region + the solidified Zn melt).

Figure 5 demonstrates a good correlation between the Zn coating (or the contribution of solidified Zn melt in the case of Figure 3f) when the growing Zn thickness obtained from the metallographic images increases along with the thickness indicated by the thickness gauge CM-8825FN. However, Figure 5 also demonstrates that the indicated values obtained from the thickness gauge CM-8825FN are higher than the ones obtained from the metallographic observations. The presence of the diffusion layer in the near-surface region of the steel matrix lying below the Zn coating, which appears white (see Figure 3f in which the diffusion layer is marked by white arrow), plays a certain role. The measurement of Zn thickness by the use of the thickness gauge CM-8825FN is based on magnetic contrast between the ferromagnetic matrix and the non-ferromagnetic coating. For this reason, the higher thickness values indicated by the thickness gauge CM-8825FN are due to the contribution of the diffusion layer and the presence of Zn in the flange matrix, which makes the diffusion layer harder from the magnetic point of view. The bigger error bars for the thickest Zn layer (as indicated in Figure 5) are due to the remarkably higher thickness variability of the Zn solidified deposition (see especially Figure 3f) as contrasted against more uniform thickness of the Zn coating originated from the galvanizing process. The chemical maps depicted in Figure 6 demonstrate the clear and sharp boundary between the Zn coating and the flange steel matrix. However, the local chemical analysis (see Figure 7) where the chemistry is investigated across the Zn layer towards the deeper steel matrix regions shows that the boundary between the Zn coating and the flange matrix is not sharp and exhibits a certain gradient (see especially Figure 7b). Detailed screening of the chemical gradients depicted in Figure 7 (especially the gradient of Zn on the boundary between Zn coating and steel matrix) reveals that the thickness of the diffusion layer is about 7 μm for the thinnest Zn coating. This finding fully explains the difference between the thickness of Zn coating obtained from LM and that measured by the thickness gauge CM-8825FN. However, thickness of the diffusion layer is 12 μm for the Zn coating of thickness 144 μm obtained from LM, whereas the Zn coating thickness measured by the thickness gauge CM-8825FN is 180 μm. Having the error bar ± 9.5 μm for this particular case, about 15 μm error of measurement should be taken into account for the thickness gauge CM-8825FN measurements. The preliminary phase of a coating thickness measurement (applying the gauge CM-8825FN) requires a calibration procedure when the measured signal is fitted against the plates of known thickness. Applying the plate of thickness 125 μm (close to the nominal thickness 135 μm), the explanation of the aforementioned measurement deviations should be linked with this aspect.

Figure 7 also confirms information obtained from the metallographic observation when the thicker Zn coating corresponds to the wider region in which Zn dominates (compare Figure 7a,b). Figure 7 also demonstrates that the Zn coating is not fully composed of Zn, but about 10% of Fe can be detected as well.

Figure 8 depicts the profile of microhardness HV0.2 after galvanizing. Microhardness in the sub-surface depth from 50 up to approx. 700 μm is higher as compared with the bulk as a result of cold forming of the flange. However, the microhardness HV0.2 in the steel matrix below the Zn coating (at the depth of about 50 μm) drops down due to thermal softening at the elevated galvanizing temperatures. Such evolution of microhardness as that illustrated in Figure 8 is nearly the same for all investigated positions without a marked difference with respect to the different Zn columnar coating.

MBN drops down along with the increasing thickness of Zn coating (see Figure 9). Note, information about the Zn coating (layer) thickness is obtained from LM employed in Figure 9, Figure 10, Figure 11, Figure 12 and Figure 13. Effective values of MBN signal in both directions exhibit good sensitivity apart from the last position for the thickest Zn layer (composed of Zn coating and deposited solidified Zn melt). However, this case can be easily detected by visual inspection, as mentioned above. Saturation of the evolution depicted in Figure 9 indicates that the attenuation of the MBN signal due to the presence of the solidified Zn melt is less as contrasted with the columnar Zn coating. Moreover, it should also be considered that the reading depth of MBN technique in this particular case [24,25] is still above the thickest Zn layer, but not far away.

Explanation of the decreasing MBN along with the increasing Zn layer is similar to that reported in [13]. The increasing thickness of Zn coating (layer) makes the magnetic field in the ferromagnetic Fe matrix of the flange weaker which slows down the speed of DWs motion and the corresponding rate of magnetization (directly connected with MBN emission [27,28]). Non-ferromagnetic Zn coating produces zero MBN emission in comparison with the paramagnetic phase and attenuates the electromagnetic pulses produced by DWs (originating from the flange) during the altering magnetising field. The degree of attenuation grows along with the increasing thickness of the Zn coating (layer).

For these reasons, the amplitude of the produced pulses drops down and the attenuation of these pulses is stronger along with the growing Zn coating (layer). MBN envelopes illustrated in Figure 10 demonstrate that the lower MBN pulses are produced at the higher magnetic fields along with increasing thickness of the Zn coating (layer), which in turn, corresponds to the higher *PP* positions (see Figure 11). Figure 11 also indicates better sensitivity of TD as contrasted with RD.

It should be considered that MBN originates from the diffusion layer and the underlying untouched matrix. The phase diagram drawn according to Schramm [29] indicates that the matrix is fully composed of the Γ phase as soon as the Zn exceeds approx. 72.5% (Fe is dissolved in the Zn matrix). Therefore, the Zn coating (as well as the solidified Zn melt) is fully paramagnetic and does not contribute to the MBN emission detected on the free surface. As soon as the Zn is in the range from 7 up to 72.5%, the bbc ferromagnetic α-phase coexists in the equilibrium with the Γ phase [30]. The contribution of the diffusion to the MBN emission in this particular case depends on the chemical gradient and the corresponding phase ratio. Finally, Zn is fully dissolved in the Fe α matrix when Zn is below 7%. For this reason, it is considered that MBN emission becomes stronger towards the deeper regions (along with the decreasing Zn dissolution) of the diffusion layer and saturates as soon as the Zn-free matrix is attained.

MicroScan software also provides information about the number of detected pulses, as depicted in Figure 12. The number of MBN pulses grows along with the increasing thickness of Zn coating (layer), whereas the MBN (effective) value of emission decreases. Such a finding is quite controversial since MBN emission is a function of the amplitude of pulses and the superimposing contribution of their number [31]. Therefore, the higher number of MBN pulses are mostly those of lower magnitude (see Figure 13), playing a minor role in the whole MBN emission expressed in the effective value of the signal. Moreover, this parameter is polluted by the floating threshold function employed in MicroScan post-processing [32].

The sensitivity of the MBN technique against the variable thickness of Zn coating (layer) can be also demonstrated when the Zn coating is etched off via an electrochemical process (see Figure 14). Figure 14 depicts the evolution of the effective value of MBN, as well as *PP,* when the thickness of Zn layer is progressively reduced. MBN progressively grows, whereas *PP* drops down when the Zn layer becomes thinner due to decreasing opposition of thinner Zn coating against magnetization, as well as the reduced length of electromagnetic pulses propagation towards the free surface. Figure 14a also demonstrates that MBN does not attain the values typical for bulk (as that in the deeper regions) when the Zn coating is fully removed due to the contribution of the diffusion layer in the near-surface region of the steel matrix. For this reason, the full saturation of MBN occurs later in the depth (when the diffusion layer is fully etched off). Figure 14b depicts that *PP* position attains the local minimum at approx. the diffusion layer due to the thermal softening originating from galvanizing at elevated temperatures. Further etching results in the moderate growth of *PP* due to the presence of the hardened matrix originating from cold forming, as indicated by the microhardness in Figure 8.

## 4. Conclusions

The main outputs of this study can be summarized as follows:−the thickness of Zn coating on the flange provided by the thickness gauge CM-8825FN is more than that obtained from LM due to the contribution of the diffusion layer,−solidification of Zn melt on the flange surface makes the thickness of Zn layer higher, but these regions can be revealed by visual inspection,−MBN decreases along with the increasing thickness of the Zn coating,−*PP*, as well as the number of MBN pulses, grows along with the increasing thickness of Zn coating.

MBN technique (especially the conventional effective values) provides very good sensitivity against the variable thickness of the Zn layer. Therefore, the magnetic measurements (Barkhausen noise as well as the thickness gauge CM-8825FN) can be employed for reliable and fast monitoring of the protective Zn coating on the steel flanges of the road barriers.

## Figures and Tables

**Figure 1 materials-15-01898-f001:**
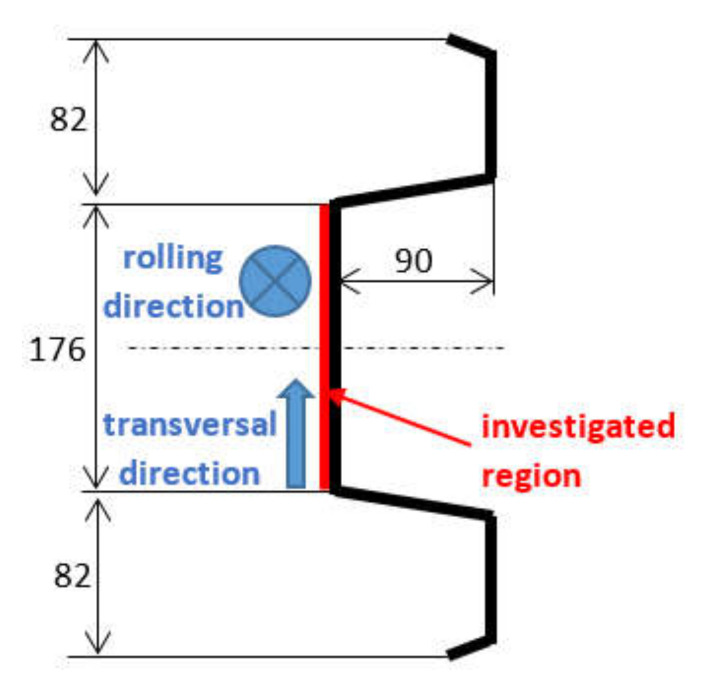
Profile of the flange NH4 with dimensions, highlighted investigated region, and the directions of MBN measurements.

**Figure 2 materials-15-01898-f002:**
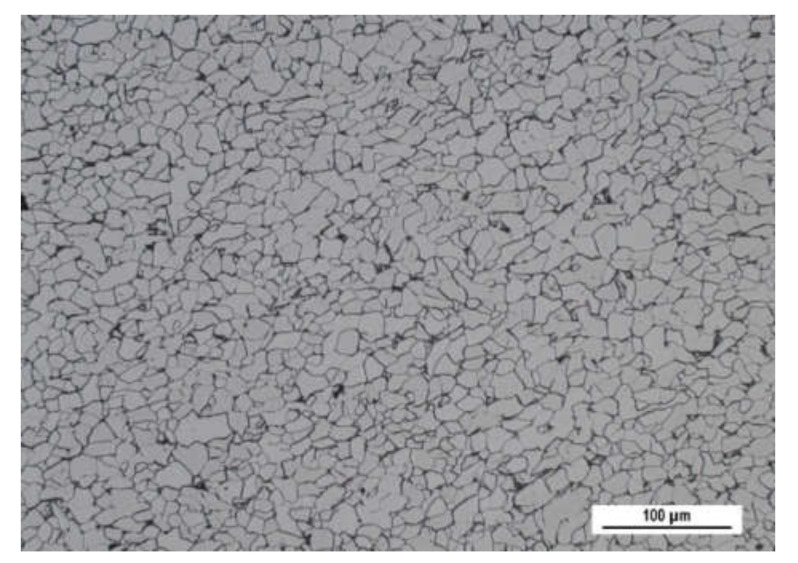
Ferrite matrix of the flange, 3% Nital.

**Figure 3 materials-15-01898-f003:**
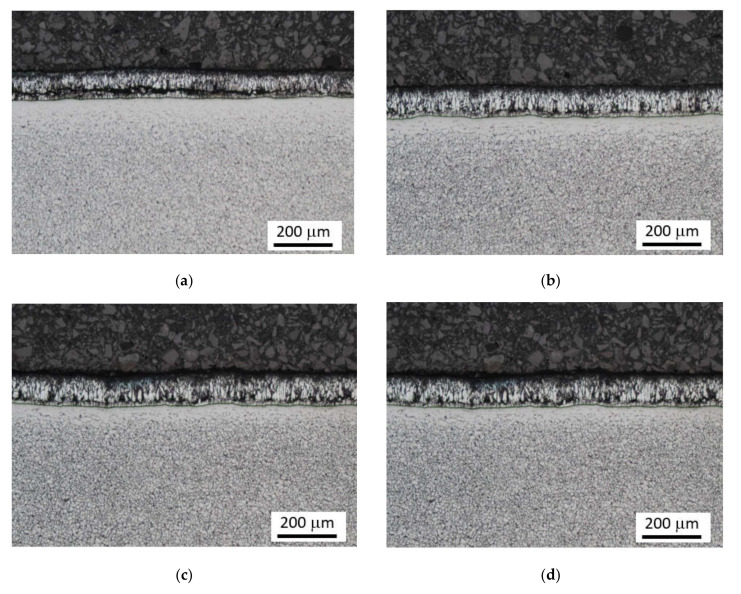
Metallographic images of Zn layers. (**a**) coating thickness 101 ± 4 μm, (**b**) coating thickness 110 ± 4 μm, (**c**) coating thickness 112 ± 3 μm, (**d**) coating thickness 127 ± 4 μm, (**e**) coating thickness 144 ± 6 μm, (**f**) coating thickness 190 ± 28 μm.

**Figure 4 materials-15-01898-f004:**
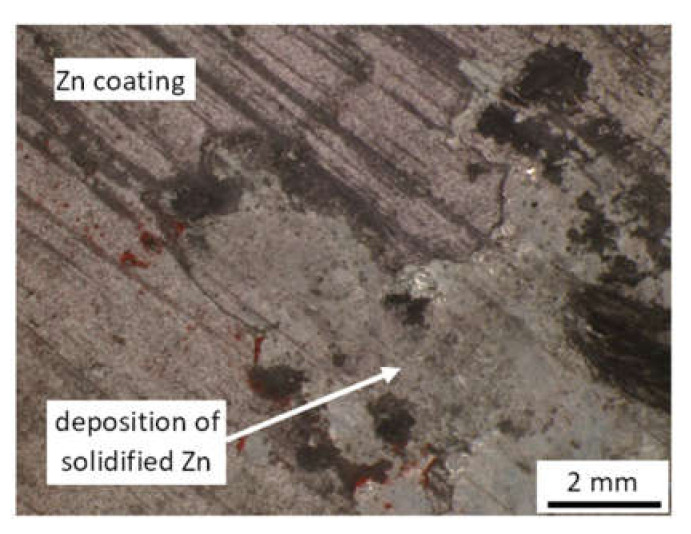
Deposition of solidified Zn melt on the Zn coating—coating thickness 190 ± 28 μm (linked with Figure 3f).

**Figure 5 materials-15-01898-f005:**
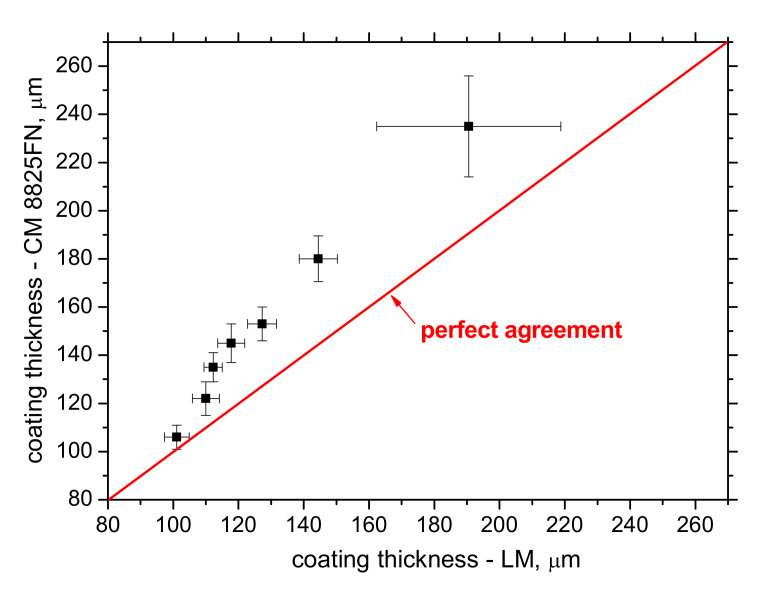
Correlation between coating thickness obtained from LM and measured by CM-8825FN.

**Figure 6 materials-15-01898-f006:**
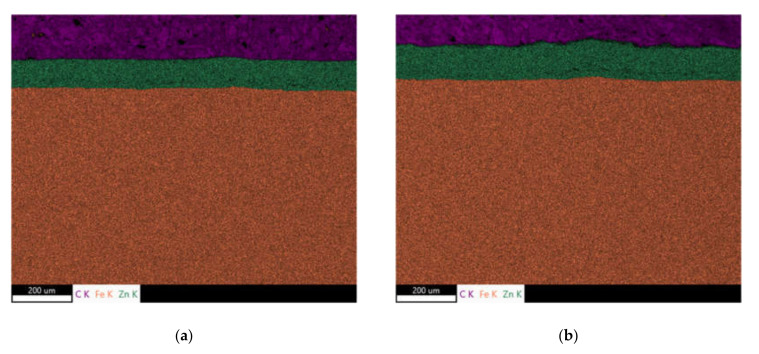
Chemical maps of Zn coating and steel matrix. (**a**) Coating thickness 101 ± 4 μm, (**b**) coating thickness 144 ± 6 μm.

**Figure 7 materials-15-01898-f007:**
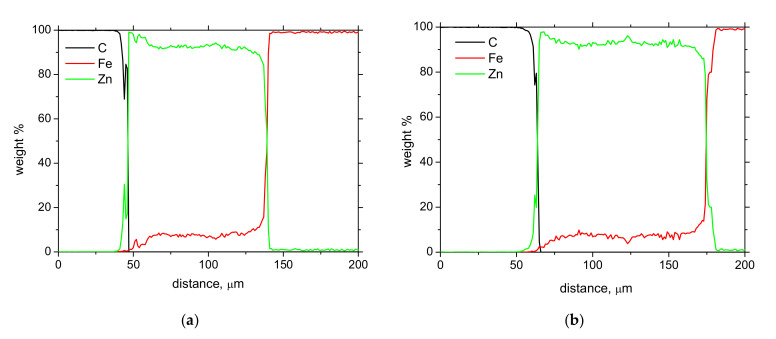
Local chemical analysis. (**a**) Coating thickness 101 ± 4 μm, (**b**) coating thickness 144 ± 6 μm.

**Figure 8 materials-15-01898-f008:**
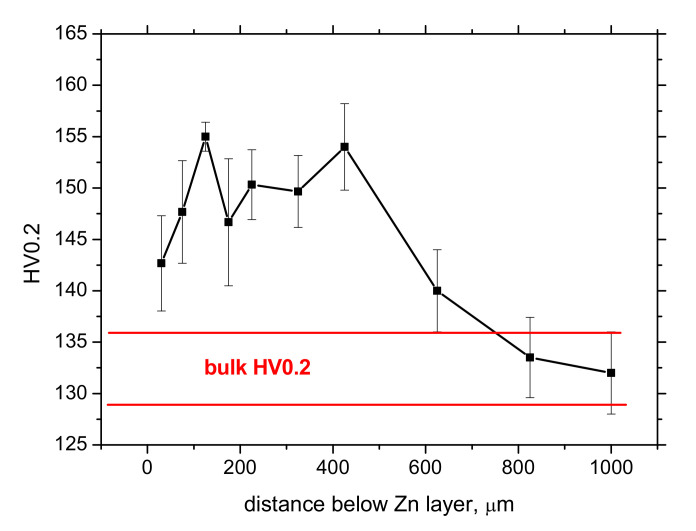
Microhardness profile HV0.2 below the Zn layer. Zero distance represents the boundary between the Zn layer and the matrix.

**Figure 9 materials-15-01898-f009:**
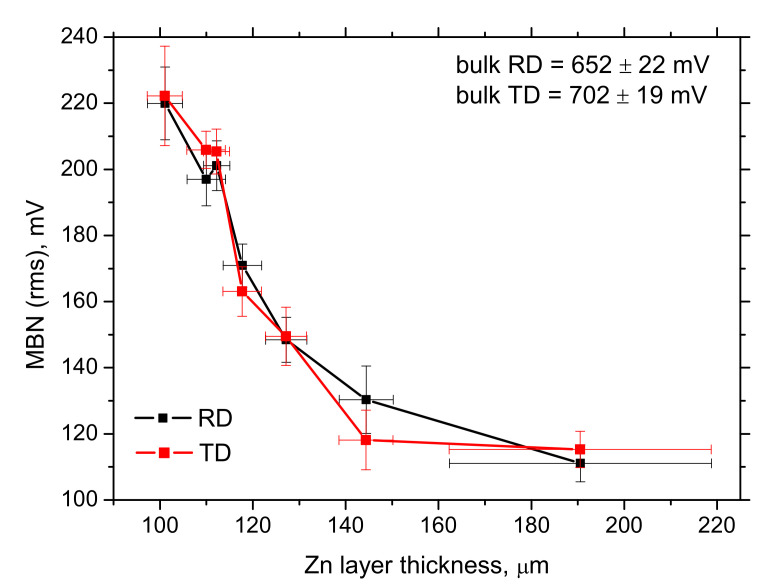
Evolution of MBN along with increasing Zn coating (layer) thickness (obtained from the MO).

**Figure 10 materials-15-01898-f010:**
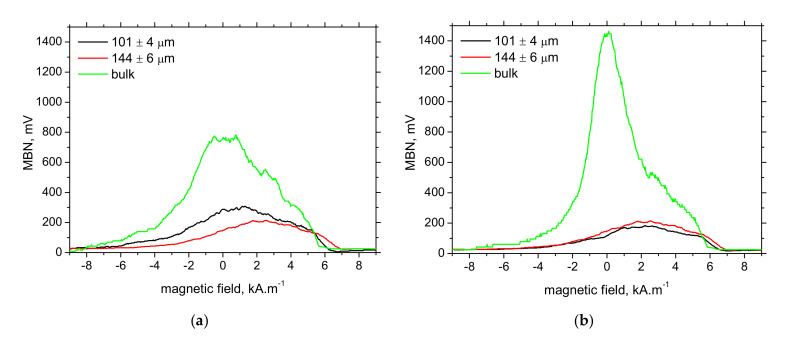
Evolution of MBN envelopes as a function of Zn coating thickness. (**a**) RD, (**b**) TD.

**Figure 11 materials-15-01898-f011:**
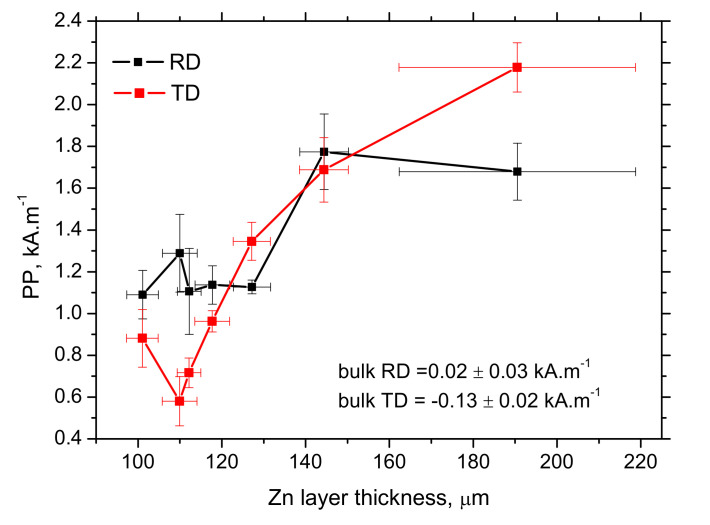
Evolution of *PP* along with increasing Zn coating (layer) thickness (obtained from the MO).

**Figure 12 materials-15-01898-f012:**
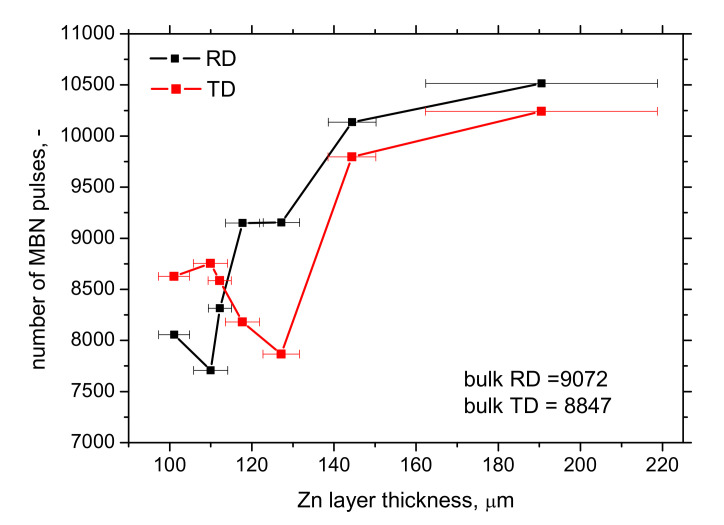
Evolution of the number of detected MBN pulses along with increasing Zn coating (layer) thickness (obtained from the MO).

**Figure 13 materials-15-01898-f013:**
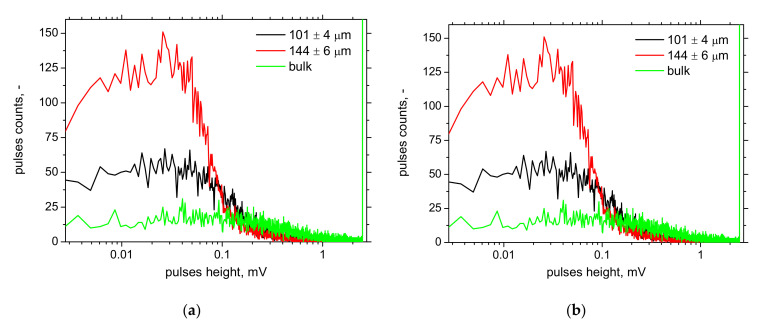
Distribution of MBN pulses height. (**a**) RD, (**b**) TD.

**Figure 14 materials-15-01898-f014:**
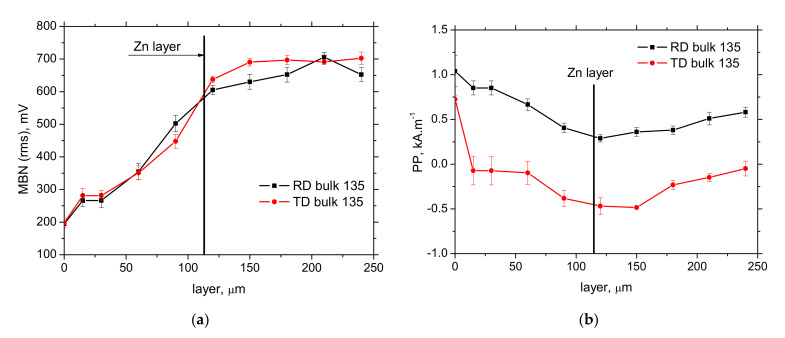
Evolution of MBN and *PP* after the consecutive removal of Zn layer (sample of the nominal thickness of Zn layer indicated by the OM 112 ± 3 μm). (**a**) MBN, (**b**) *PP*.

**Table 1 materials-15-01898-t001:** Chemical composition of the flange steel matrix in wt.%.

Fe	C	Mn	Si	Cr	Co	Al	Cu	Ni
balance	0.293	0.497	0.185	0.036	0.010	0.080	0.076	0.017

## Data Availability

Data available on any request.

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
