# Peer review of "Magnetic Measurement of Zn Layer Heterogeneity on the Flange of the Steel Road Barrier"

_materials, 2022, doi:10.3390/ma15051898_

Round 1

Reviewer 1 Report

The authors applied combined magnetic measurements (Barkhausen noise and thickness gauge CM-8825FN) to monitor Zn coating thickness of steel flanges of road barriers. This may be exciting due to its reliability and efficiency. But I have major concerns with the originality (as written) of this paper and some questions on the measured results.

A series of similar papers from the same authors have been noticed. Some most recent ones are:

Zgútová, Katarína, and Martin Pitoňák. "Attenuation of Barkhausen Noise Emission due to Variable Coating Thickness." Coatings 11.3 (2021): 263.

Jančula, Miroslav, et al. "Monitoring of Corrosion Extent in Steel S460MC by the Use of Magnetic Barkhausen Noise Emission." Journal of Nondestructive Evaluation 40.3 (2021): 1-11.

The authors need to justify the significance of the present manuscript that makes it distinct from these previous works, in terms of the originality.

In this manuscript, the coating thickness measured by the thickness gauge is consistently larger than that from direct metallographic observations (MOs). The authors explained this based on diffusion. While part of me think this sounds possible, several doubts have diverged my thoughts from this:

  1. In Figure 5, as the coating thickness gets larger, the deviation between the coating thickness from the gauge and that from the MO increases. Solely the diffusion does not explain this behavior.
  2. A relevant question: The authors should explain why the error bars become bigger as the coating thickness increases.
  3. I can see from Figure 7 that there is some diffusion profile across the phase boundary, but unfortunately, these diffusion layers look only several micron-thick, much smaller than the deviations shown in Figure 5, especially as the coating thickness gets bigger. The authors should articulate this.

Author Response

Reviewer: But I have major concerns with the originality (as written) of this paper and some questions on the measured results. A series of similar papers from the same authors have been noticed. Some most recent ones are:

Zgútová, Katarína, and Martin Pitoňák. "Attenuation of Barkhausen Noise Emission due to Variable Coating Thickness." Coatings 11.3 (2021): 263.

Jančula, Miroslav, et al. "Monitoring of Corrosion Extent in Steel S460MC by the Use of Magnetic Barkhausen Noise Emission." Journal of Nondestructive Evaluation 40.3 (2021): 1-11.

The authors need to justify the significance of the present manuscript that makes it distinct from these previous works, in terms of the originality.

Response: We agree. We added further explanation.

Manuscript: As compared with the previous studies [7, 20] focused on presence of coating or corrosion on the ferromagnetic matrix, this study also reports about MBN attenuation as a result of diffusion layer laying below the non-ferromagnetic coating. Zn chemistry gradient at the boundary between Zn coating and the steel matrix dims the contrast between ferromagnetic and paramagnetic phases. Also contribution of the Zn melt solidification after galvanizing is investigated as well. 

Reviewer: In Figure 5, as the coating thickness gets larger, the deviation between the coating thickness from the gauge and that from the MO increases. Solely the diffusion does not explain this behavior.

Reviewer: I can see from Figure 7 that there is some diffusion profile across the phase boundary, but unfortunately, these diffusion layers look only several micron-thick, much smaller than the deviations shown in Figure 5, especially as the coating thickness gets bigger. The authors should articulate this.

Response: We fully agree with the reviewer. These comments are very reasonable. 

Manuscript: We added this text.

Detail screening of the chemical gradients depicted in Figure 7 (especially the gradient of Zn on the boundary between Zn coating and steel matrix) reveals that thickness of the diffusion layer is about 7 mm for the thinnest Zn coating. This finding fully explains the difference between the thickness of Zn coating obtained from LM and that measured by the thickness gauge CM-8825FN. However, thickness of the diffusion layer is 12 mm for the Zn coating of thickness 144 mm obtained from LM whereas the Zn coating thickness measured by the thickness gauge CM-8825FN is 180 mm. Having the error bar ± 9.5 mm for this particular case, about 15 mm error of measurement should be taken into account for the thickness gauge CM-8825FN measurements. The preliminary phase of a coating thickness measurement (applying the gauge CM-8825FN) requires calibration procedure when the measured signal is fitted against the plates of known thickness. Applying the plate of thickness 125 mm (close to the nominal thickness 135 mm), the explanation of the aforementioned measurement deviations should be linked with this aspect. 

Reviewer: A relevant question: The authors should explain why the error bars become bigger as the coating thickness increases.

Response: Error bars for all coating thickness are quite similar apart from the thickest one which is mixture of Zn columnar coating and deposition of Zn melt on the Zn coating since the thickness of solidified layer alters remarkably. Please check Fig. 3f.

Manuscript: We added comment.

The bigger error bars for the thickest Zn layer (as that indicated in Figure 5) are due to the remarkably higher thickness variability of the Zn solidified deposition (see especially Figure 3f) as contrasted against more uniform thickness of the Zn coating originated from galvanising process. 

Reviewer 2 Report

Manuscript title: Magnetic measurements of Zn layer heterogeneity on the flange of the steel road barrier

The manuscript deals with the magnetic measurements of the Zn as a protective covering layer of the steel road barriers providing a corrosion resistance process in road barriers against the surrounding environmental conditions. This topic is imperative in a wide span of applications and daily activities. The manuscript in general is decent, and qualified for publication after addressing some minor comments

  • In the abstract, the most significant attained results should be mentioned.
  • The introduction should highlight more the background of the work and the function of the Zn protective layer with recently published works.
  • Pg.2, Table 1: revise the sum of the wt% ( higher).
  • Pg.2, figure 1: the figure should provide full details and explanation along with the caption without the need to refer to the text, what are the numbers and abbreviations?
  • Pg.4, line 104: on the other hand>> conversely
  • Pg.6 Fig. 7(a) distance.,>>distance,
  • Pg.7, line 152: the figures 9÷13>>figures 9-13
  • Pg.7, line 165: For these reasons, amplitude>> For these reasons, the amplitude..
  • Pg.8 Figure 10: better to unify y-axis scale in (a) and (b). and provide full details in the caption of all figures as stated earlier.
  • Pg.9, line 187: Contribution of the diffusion to>> The Contribution of the diffusion to..
  • Pg. 10, line 200: playing the minor role>> playing a minor role
  • Pg.1010, line 204: depicts evolution>> depicts the evolution
  • Pg.1010, fig.14: ZN>>Zn

Best of luck

Author Response

Reviewer: In the abstract, the most significant attained results should be mentioned.

Response: We gently enlarged conclusions.

Manuscript: We added these sentences.

Number of strong MBN pulses drops down with the increasing thickness of Zn coating at the expense of increasing number of the weak MBN pulses. The thickness of Zn coating can be polluted by the solidification of Zn melt after galvanizing. Presence of the diffusion layer dims the contrast between ferromagnetic and paramagnetic phases.

Reviewer: The introduction should highlight more the background of the work and the function of the Zn protective layer with recently published works.

Response: We added the required information.

Manuscript:

Zn galvanizing process is widely investigated and new regimes with respect of galvanizing temperature or/and chemistry of bath are proposed with respect of their functionality. Kania et al. [6] reported about the negative influence of Bi added to a Zn bath on the corrosion resistance due to the presence of Bi precipitates. Li et al. [7] remarkably improved adhesion of Zn coating after the cold galvanizing by the use of γ-chloropropyl triethoxysilane. Cetinkaya et. all [8] reported about the impact of chemical treatment on Zn-Al-Mg coating produced by a hot-dip process in order to increase Zn fraction at the coating surface. The influence of galvanizing temperature on corrosion resistance was also investigated [9] as well as the contribution of Mn to the morphology and corrosion resistance of Zn coating [10]. Finally, Kang et al. [11] investigated Fe-Zn reaction during a hot-dip galvanizing and its influence on microcracking. The aforementioned brief list of the latest findings in this field indicates that Zn coating technology is widely investigated and many improvements have been proposed in order to improve the corrosion resistance of bodies. However, a fast, reliable and especially non-destructive technique (for their monitoring) being sensitive to the thickness of the Zn layer could be also beneficial.   

we added these papers into the list of references

  1. Kania, H.; Saternus, M.; Kudláček, J.; Svoboda, J. Microstructure Characterization and Corrosion Resistance of Zinc Coating Obtained in a Zn-AlNiBi Galvanizing Bath. 2020, 10, 758; doi: 10.3390/coatings10080758.
  2. Li, J. et all. Interface Characteristics and Anticorrosion Performances of Cold Galvanizing Coatings Incorporated with γ-chloropropyl Triethoxysilane on Hot-Dip Galvanized Steel. 2021, 11, 402; doi: 10.3390/coatings11040402.
  3. Cetinkaya, B.W.; Junge, F.; Muller, G.; Haakmann, F.; Schierbaum, K.; Giza, M. Impact of alkaline and acid treatment on the surface chemistry of a hot-dip galvanized ZneAleMg coating. Mater. Res. Technol. 2020, 9, 16445-16458; doi: 10.1016/j.jmrt.2020.11.070.
  4. Kancharla, H.; Mandal, G.K.; Singh, S.S.; Mondal, K. Effect of strip entry temperature on the interfacial layer and corrosion behavior of galvanized steel. Coat. Technol. 2022, 433, 128071; doi: 10.1016/j.surfcoat.2021.128071
  5. Grandhi, S.; Raja, V.S.; Parida, S. Effect of manganese addition on the appearance, morphology, and corrosion resistance of hot-dip galvanized zinc coating. Coat. Technol. 2021, 421, 127377; doi: 10.1016/j.surfcoat.2021.127377.
  6. Kang, J.H.; Kim, D.; Kim, D.H.; Kim, S.J. Fe-Zn reaction and its influence on microcracks during hot tensile deformation of galvanized 22MnB5 steel. Coat. Technol. 2019, 357, 1069-1075; doi: 10.1016/j.surfcoat.2018.08.010

Reviewer: Pg.2, Table 1: revise the sum of the wt% ( higher).

Response: It seems being correct since the fraction of alloying elements wt% are indicated and the rest of the body is Fe indicated as balanced. 

Manuscript: We prefer no change.

Reviewer: Pg.2, figure 1: the figure should provide full details and explanation along with the caption without the need to refer to the text, what are the numbers and abbreviations?

Response: We altered the appearance of Fig. 1.

Manuscript: please check Fig. 1.

Reviewer: Pg.4, line 104: on the other hand>> conversely

Response: corrected

Manuscript: Conversely, these isolated regions…

Reviewer: Pg.6 Fig. 7(a) distance.,>>distance,

Response: there is no dot.

Manuscript: no change

Reviewer: Pg.7, line 152: the figures 9÷13>>figures 9-13

Response: corrected

Manuscript: …in the Figures 9-13.

Reviewer: Pg.7, line 165: For these reasons, amplitude>> For these reasons, the amplitude..

Response: corrected

Manuscript: For these reasons, the amplitude…

Reviewer: Pg.8 Figure 10: better to unify y-axis scale in (a) and (b). and provide full details in the caption of all figures as stated earlier.

Response: we unified Y-axis and added details in the caption.

Manuscript: Please check Fig. 10.

Reviewer: Pg.9, line 187: Contribution of the diffusion to>> The Contribution of the diffusion to..

Response: corrected

Manuscript: The contribution of the diffusion to the…

Reviewer: Pg. 10, line 200: playing the minor role>> playing a minor role

Response: corrected

Manuscript: …playing a minor role…

Reviewer: Pg.1010, line 204: depicts evolution>> depicts the evolution

Response: corrected

Manuscript: Figure 14 depicts the evolution …

Reviewer: Pg.1010, fig.14: ZN>>Zn

Response: corrected

Manuscript: please check the appearance of Fig. 14.

Round 2

Reviewer 1 Report

The authors have provided detailed explanations in the response letter and the modified manuscript.